# Magnetron Sputtering of Transition Metals as an Alternative Production Means for Antibacterial Surfaces

**DOI:** 10.3390/microorganisms10091843

**Published:** 2022-09-15

**Authors:** Bernhard Peter Kaltschmidt, Ehsan Asghari, Annika Kiel, Julian Cremer, Dario Anselmetti, Christian Kaltschmidt, Barbara Kaltschmidt, Andreas Hütten

**Affiliations:** 1Department of Thin Films and Physics of Nanostructures, Center of Spinelectronic Materials and Devices, Faculty of Physics, Bielefeld University, 33615 Bielefeld, Germany; 2Department of Cell Biology, Faculty of Biology, Bielefeld University, 33615 Bielefeld, Germany; 3Department of Experimental Biophysics & Applied Nanosciences, Faculty of Physics, Bielefeld University, 33615 Bielefeld, Germany

**Keywords:** magnetron sputtering, transition metals, biofilm, antibacterial surfaces, pseudomonas aeruginosa

## Abstract

In the light of the SARS-CoV-2 pandemic and growing numbers of bacteria with resistance to antibiotics, the development of antimicrobial coatings is rising worldwide. Inorganic coatings are attractive because of low environmental leakage and wear resistance. Examples for coatings are hot metal dipping or physical vapor deposition of nanometer coatings. Here, magnetron sputtering of various transition metals, such as gold, ruthenium and tantalum, was investigated. Metal films were characterized by scanning electron microscopy (SEM), atomic force microscopy (AFM) and energy dispersive X-ray spectroscopy (EDX). We investigated the growth of Pseudomonas aeruginosa isolated from household appliances on different sputter-coated metal surfaces. The fine-grained nanometric structure of these metal coatings was between 14 nm (tantalum) and 26 nm (gold) and the roughness was in a range of 164 pm (ruthenium) to 246 pm (gold). Antibacterial efficacy of metal surfaces followed the order: gold > tantalum > ruthenium. Interestingly, gold had the strongest inhibitory effect on bacterial growth, as analyzed by LIVE/DEAD and CFU assay. High-magnification SEM images showed dead bacteria characterized by shrinkage induced by metal coatings. We conclude that sputtering might be a new application for the development of antimicrobial surfaces on household appliances and or surgical instruments.

## 1. Introduction

Since 2019, the global SARS-CoV-2 pandemic has shocked the world with several million deaths and the following economic collapse. Thus, interest has increased towards the development of advanced antibacterial and antiviral surfaces. The global antimicrobial coatings market size is projected to grow from USD 3.9 billion in 2021 to USD 6.4 billion by 2026, with an annual growth rate of 10.5% [1].

Physical vapor deposition (PVD) is a method frequently used to generate thin films composed of metals or metal oxides. This method has been used for decades and significant improvements have been made in the quality of coatings and speed of deposition rate [2,3]. The term PVD appeared in the 1960s, since technology evolution led to the invention of new physical tools, such as plasma technology, high-vacuum technology, electro-magnetic acceleration of particles and the development of magnetron sputtering [4]. The magnetron is a physical device using magnetic fields to improve plasma density during the sputtering process. For sputtering, a magnetron is positioned near the metallic target (Au, Ta, Ru). An ionic gas is formed by strong electric fields and accelerated to the target, thus, releasing atomic-sized metal particles, which are deposited onto the glass surface (sputtering substrate) [5]. Many coating techniques have been used to generate antimicrobial surfaces [6]. Of all metal coating methods, magnetron sputtering appears superior, since it generates highly adhesive and flexible thin films [7], which are very wear resistant and can be used on temperature-sensitive objects [6]. So far, magnetron sputtering is not heavily applied for antiviral and antibacterial surfaces, since it demands high-vacuum technology. Rare examples of magnetron sputtering for antimicrobial surfaces include TiO2 in different phases (anatase and rutile) [8] or different multi layers, such as AgCu [6,9] or TiCu [10]. Magnetron sputtering was previously not used for gold as a single layer for antibacterial surfaces [6,11].

In our study, we tried to develop new coatings in order to protect household appliances from microbial contamination, see Figure 1. Biofilms are frequently detected in home environments [12] and household appliances, such as washing machines [13,14], dishwashers [15] and coffee makers [16]. Here, we produced coated glass surfaces using magnetron sputtering of different transition metals. The next step was to analyze the coatings of gold (Au), ruthenium (Ru) and tantalum (Ta) for biofilm formation and compare the results to uncoated glass surfaces. We used our previously isolated wild-type Pseudomonas aeruginosa strain, cultured from household appliances [17]. We observed that this specific wild-type P. aeruginosa is a very strong biofilm producer compared to other bacteria obtained from strain collection [17]. This type of P. aeruginosa is especially hard to kill and rapidly forms biofilms on various surfaces.

Here, we used this bacterium to analyze the antibacterial effect on sputter-coated surfaces. To detect bacterial cell death, several assays were used, such as scanning electron microscopy (SEM), LIVE/DEAD assays with confocal laser scanning microscopy (CLSM) and colony forming unit (CFU) assays. This was depicted vividly as “happy” bacteria on glass bacterial growth, whereas coating with transition metals is lethal and leads to dying bacteria, depicted as “unhappy” bacteria. Strongest growth inhibition was observed on gold and tantalum sputter-coated surfaces. These observations might pave the way for magnetron sputter-coated gold surfaces of household appliances and/or surgical instruments to avoid unhealthy bacterial growth.

The results of this study could be translated to the use of innovative antibacterial surfaces, which might lead to a lower consumption of antimicrobial chemicals in household appliances, thus, protecting the environment.

## 2. Materials and Methods

### 2.1. Sputter Coating

Glass slides with a size of 22 mm × 22 mm were sputter coated by the use of a self-built magnetron sputtering device. The glass slides were cleaned with 70% ethanol followed by nitrogen drying. The sputtering conditions were as follows: deposition temperature was 20 °C, chamber pressure was 1 × 10^−6^ mbar and the sputtering power was 25 Watt (DC mode). During the sputter process the chamber pressure was 3.5 × 10^−3^ mbar after the application of Argon as sputtering gas. The materials used for sputter coating were gold, ruthenium and tantalum targets with a diameter of 32 mm and the distance to the substrate was 2 cm. A coating thickness of 20 nm was chosen. Because the adhesion of gold to the glass surface was not sufficient a layer of 10 nm ruthenium was applied first to act as a bonding agent, though only for the gold coating.

### 2.2. Bacterial Species

The strong biofilm producer Pseudomonas aeruginosa, isolated from a domestic washing machine [17], was used for all experiments. The bacterial isolate was inoculated from frozen stocks kept under −80 °C and cultured for 24 h on LB agar plates.

### 2.3. Atomic Force Microscopy

To gain information about the surface topography AFM surface roughness measurements were used. The AFM measurements were conducted with a NanoWizard™ UltraSpeed 2 (JPK-Bruker, Berlin, Germany). The data were recorded using a Tap300Al-G cantilever from BudgetSensors^®^ (Innovative Solution Bulgaria Ltd., Sofia, Bulgaria) operating in tapping mode. The data were evaluated with the open-source software Gwyddion [18].

### 2.4. Scanning Electron Microscopy and Energy Dispersive Elemental Analysis

For an ultrastructural observation of biofilms grown on sputter-coated glass slides, we used scanning electron microscopy (SEM). P. aeruginosa was inoculated from fresh LB agar plates and pre-cultured in 10 mL LB medium overnight. The glass samples were incubated in bacterial solution adjusted at OD600 = 0.01 for 24 h at 37 °C in static conditions.

After the incubation period, planktonic cells were washed away by submerging the samples twice in physiological saline (0.9% NaCl) followed by once in bidest H2O. The samples were fixed with half-strength Karnovsky’s solution (2% paraformaldehyde, 2.5% glutaraldehyde) for 30 min. The fixed samples were dehydrated using 50, 70, 80, 90, 95 and 100% (*v/v*) graded ethanol followed by t-butyl alcohol. To improve the conductivity of the samples, they were sputter coated with a layer of 4 nm Ruthenium. For examination, a Helios NanoLab DualBeam 600 (FEI Company, Hillsboro, OR, USA) scanning electron microscope (SEM) was used. For image acquisition, the microscope parameters were set to an acceleration voltage of 5 kV and a beam current of 0.17 nA. To ensure conductivity and reduce charging effects the coated surfaces were connected to the stage by the use of conductive carbon tape.

Elemental analysis of the sputter-coated surfaces was carried out by an EDAX Apollo 10 (EDAX company, Mahwah, NJ, USA) energy dispersive X-ray (EDX) detector. To ensure a proper signal during EDX analysis acceleration voltage and beam current were increased to 20 kV and 0.34 nA, respectively.

The size measurement of the grain structure was performed by using the open-source software FIJI (ImageJ).

### 2.5. Confocal Laser Scanning Microscopy

To determine living and dead bacteria and localization of potential biofilm, LIVE/DEAD staining followed by confocal laser scanning microscopy (CLSM) was used. The samples were cultured in the same way as described in the SEM section. The FilmTracer™ LIVE/DEAD Biofilm Viability Kit (Molecular Probes, Invitrogen, Carlsbad, CA, USA) was used according to the manufacturer’s instructions. The SYTO 9 green fluorescent nucleic acid stain measured at 482 nm excitation and 500 nm emission was used to visualize live bacteria with an intact cell membrane. Bacteria with a compromised membrane, considered dead or dying, were stained with propidium iodide (red), measured at 490 nm excitation and 635 nm emission. To obtain a percentage of live and dead cells, five images of the same magnification were evaluated with FIJI (ImageJ) software for each surface using the following routine: First, the channels (live and dead) were split. The images were then converted into binary images. Then the amount of white and black pixels for each image was calculated with the FIJI selection tool. The last step was to compare the number of pixels of the living cells to the number of pixels of the dead cells. Results are presented in percent.

Additionally, to measure reactive oxygen species we used the 2-7-dichlorofluorescin diacetate (DCFDA or H2DCFDA) followed by CLSM. The DCFDA/H2DCFDA-Cellular ROS Assay Kit (Abcam, Berlin, Germany) was used according to the manufacturer’s guideline. Again, samples were cultivated as described in the SEM section. DCFDA is a membrane-permeable dye that can be used to measure the activity of reactive oxygen species (ROS) in living cells. DCFDA can be measured at 485 nm excitation and 535 nm emission. To obtain a percentage from the highest to lowest activity of reactive oxygen species, three images for each surface with the same magnification were evaluated using FIJI (ImageJ). First, the images were converted to 8 bit. Then the Image J Plugin fire was used to create false color coding. High ROS intensities are displayed in white to yellow. Background and low intensities displayed in black to blue. Next, the Image J Plugin Histogram was used and pixel binning of 0–63, 64–127, 128–191 and 192–255 was performed. The images were always adjusted with the background set to black (value 0) without any bacteria.

### 2.6. Colony Forming Unit Assay

To test the antibacterial activity of the different sputter coatings in terms of viable colonies, the colony forming unit assay was performed as a direct quantification method. To form biofilms on a distinct surface area, clone cylinders (Carl Roth GmbH + Co. KG, Karlsruhe, Germany) with a diameter of 1.2 cm were placed on top of the sputter-coated glass slides. In order to prevent leaking out, the clone cylinders were fixed with silicone grease (Elkem, Oslo, Norway). Therefore, P. aeruginosa was inoculated from fresh LB agar plates and pre-cultured in 10 mL LB medium overnight. Three samples from each sputter coating were inoculated with 750 μL bacterial suspension (inside the cloning cylinder), adjusted at OD600 = 0.01. The samples were placed inside Petri dishes (Carl Roth GmbG + Co. KG, Karlsruhe, Germany) with a diameter of 6 cm and incubated for 24 h at 37 °C in static conditions. A sterility control, containing only LB medium, was additionally performed. Afterwards, planktonic cells were washed away by submerging the samples three times in physiological saline (0.9% NaCl). The biofilm was detached from the surface by vigorous vortexing for at least 1 min (Vortex Genie, Fisher Scientific, MA, USA) in 10 mL physiological saline (0.9% NaCl). A serial dilution series was performed and 100 μL of each dilution was plated onto fresh LB agar plates. The plates were incubated overnight at 37 °C. After incubation, colonies were counted and the colony forming units per mL were calculated.

### 2.7. Statistical Analysis

For all statistical analysis an unpaired nonparametric Mann–Whitney rank test was performed with *p* value output style GP with: 0.1234 (ns), 0.0332 (*), 0.0021 (**), 0.0002 (***) and <0.0001 (****). Definition of statistical significance: *p* < 0.05 (*), *p* < 0.01 (**), *p* < 0.001 (***), *p* < 0.0001 (****). Error bars always represent the standard error of mean (SEM) in all experiments.

## 3. Results

Coating with transition metals (gold (Au), ruthenium (Ru) and tantal (Ta)) was performed by magnetron sputtering. To gain information about the elemental composition of coated and uncoated glass slides, EDX measurements were performed (see Figure 2A–D).

The EDX spectrum of glass (Figure 2D) reveals that the glass sample not only consists of silicon and oxygen, but also contains traces of boron, natrium, aluminum, kalium and barium. These peaks corresponding to the glass sample can be seen in the spectra of all transition metals, because the here-analyzed coatings are relatively thin (20 nm) and the X-rays used for analysis are generated in a depth up to several micrometers. In the gold EDX spectrum (Figure 2A), a ruthenium peak can also be seen. This is due to the fact that we used a 10 nm ruthenium layer below the gold coating to act as a bonding agent. Because the energy peaks of SiKα (1.739 keV) and TaMα (1.709 keV) are very close together, the silicon peak overshadows the tantalum peak (see Figure 2C). Interestingly, a high oxygen peak can be seen for the uncoated glass sample, which is higher than the oxygen peaks observed for the sputter-coated glass samples. This might be due to the cleaning process of the samples. All samples were cleaned with ethanol followed by nitrogen drying before sputtering and microscopical analysis. The sputter-coated samples were exposed to a longer vacuum in the sputtering facility. This might explain why the ethanol residues on the sample, which can occur during the cleaning process, largely disappeared. The glass sample was placed in the microscope directly after the cleaning process.

In addition, we characterized the nanostructure of the coatings by SEM and AFM (see Figure 2E–G and Figure 3). All images showed that the morphology of the coatings was homogenous, with the exception of some larger clusters. They showed a fine-grained nanostructure, with an average grain size of 26 ± 5 nm for the gold surface and an average grain size of 17 ± 4 nm for the ruthenium surface, both in SEM and AFM images. The grain size of the tantalum surface was too small to be analyzed by SEM (see Figure 2G), but AFM measurements yielded a grain size of 14.5 ± 3 nm.

Furthermore, we analyzed the surface roughnesses using high-resolution AFM (see Figure 3). For a better visualization, we colored the glass surface in blue (Figure 3A), the gold surface in a gold tone (Figure 3B), the ruthenium surface in silver (Figure 3C) and the tantal surface in grayish green (Figure 3D). In comparison to the roughness of the uncoated glass control (RRMS 174 pm), roughness increased when coated with Au (RRMS 246 pm). For the two other transition metals (Ru: 164 pm and Ta: 179 pm), the roughness was nearly identical to glass control. Taken together, all used surfaces have a comparatively low roughness.

After characterization of the roughness, microbial colonization and biofilm formation on these metal-coated surfaces were analyzed by scanning electron microscopy (SEM). Overview pictures of different coatings are shown in Figure 4A–D. A great variation in colonization was observed. Sputtered gold surfaces (Figure 4B) revealed nearly no bacterial growth, in contrast to uncoated glass (Figure 4A) and ruthenium- and tantalum-coated surfaces (Figure 4C,D). The lowest amount of biofilm in the form of isolated insulae was visible when the surface was coated with tantalum. However, ruthenium coating induced the formation of mesh-like biofilms, in contrast to a dense nearly perfect biofilm on the uncoated glass surface. Higher-magnification SEM images (10,000×) demonstrate the formation of a dense biofilm on glass and sparsely organized biofilms on ruthenium- and tantalum-coated surfaces. In contrast, gold-coated surfaces showed only isolated bacteria and bacterial debris (Figure 4F). Highest-magnification SEM images (50,000×) revealed a healthy biofilm, where bacteria are interconnected with nanotubes on uncoated glass. As an example, for bacterial growth, a dividing bacterium is shown (see Figure 4I black arrows). Nearly intact bacteria were seen on ruthenium-coated samples (see Figure 4K black arrow), whereas dead bacteria (see Figure 4K white arrow) and abundant slime production were visible on samples coated with tantalum (see Figure 4L, white arrowhead). In a high-resolution image, a dead P. a. bacterium can be seen on a gold-coated surface, as indicated by cell wall destruction (see Figure 4J). The collapsed hull of a lysed bacterium is clearly visible.

To assess bacterial viability, we analyzed living and dead bacteria by confocal laser scanning microscopy (CLSM) using a live dead assay. Living bacteria are seen in green, while dead bacteria are stained in purple. On glass, bacterial survival was higher compared to the metal-coated surfaces (Figure 5). Au- and Ta-coated surfaces depicted a slight increase in the number of dead cells in the comparison of live cells (Figure 5B,D,F,H), whereas live cells were prevalent on glass- and Ru-coated surfaces (Figure 5A,C,E,G). Despite significant differences in the whole amount of bacteria in biofilms, all biofilms are composed of living and dead bacteria.

To measure viable bacteria attached to different surfaces in a quantitative manner, a colony forming unit (CFU) assay was performed. The areas were confined by cloning cylinders to ensure the same measurement area for every sample (see Figure 6A). Using CFU assays, a negative control with medium only on glass surfaces revealed no colony growth. In contrast, colony growth of P. a. on glass was highly effective (around 1.3 × 10^8^). Interestingly, coating with gold extremely reduced colony growth (Au: 1 × 10^7^). In the same range, tantalum reduced microbial growth, whereas Ru was less efficient (Ta: 1.5 × 10^7^ vs. Ru: 3.8 × 10^7^).

To discover potential mechanisms for cell death induced by different metal coatings, we analyzed the production of reactive oxygen species (ROS) (Figure 7A–D). ROS production is evident in all samples. Sputter coating with gold induced the highest amount of ROS production. ROS production is depicted as a false color image (Figure 7) to better visualize the different intensity values. Highest values are presented as white/yellow pixels, whereas lowest intensity is color coded in blue/black. Analysis with binning of 256 pixel values is shown in Figure 7E. Here, sputter coating with gold generates clearly visible high-intensity pixels (yellow and orange), whereas tantalum shows a lower percentage of high-intensity pixels and glass and ruthenium show the lowest. This data analysis points to high ROS production catalyzed by the nanostructured gold surface.

## 4. Discussion

Here, we analyzed the effect of transition metals gold, ruthenium and tantalum on bacterial growth. It is well known that, for example, silver and gold have high antimicrobial activity in comparison to other noble metals. Since ancient times, materials, such as silver, gold and copper, have been used as germ-destroying agents to preserve water and food [19,20]. These materials can be applied as pure metals, as metal oxides or in combination with polymers [9,21,22,23,24,25,26,27,28].

New technologies are necessary to combat bacteria, often multi resistant to antibiotics, in our environment. In a recent study, we analyzed the potential antimicrobial property of copper nanoparticle-infused polylactic acid [29]. Interestingly, we could not show any growth inhibition with the bacterium used in both studies, a wild-type Pseudomonas aeruginosa, isolated from washing machines. In contrast, P. a. even showed stress-induced growth in the presence of copper nanoparticles. In this study, we analyzed sputtered metal surfaces and not-synthesized nanoparticles and selected different metals. Surprisingly, wild-type P. a. was sensitive to gold-, ruthenium- and tantal-coated surfaces, as shown by the reduction in viable colonies in CFU assays. In addition to CFU, we used different methods for analysis of bacterial viability, such as SEM- and CLSM-based LIVE/DEAD assay. SEM analysis revealed destruction of bacterial cell walls.

Magnetron sputtering was used to apply thin layers of metals on glass. While many coating techniques have been applied to obtain antibacterial surfaces, most metal coating techniques use high temperatures and are not useful for temperature-sensitive materials, such as polymers. A new trend in household washing machines is the use of reinforced polymers for even standard metal parts, such as a drum in a washing machine. Even large-scale industrial processes, such as coating glasses for sunglasses, are used by several companies, such as Saint-Gobain [30]. Previous studies have shown an antibacterial effect of sputter-coated layers from copper and silver [6,31,32]. Czerwińska-Główka and coworkers analyzed the antibacterial effect of sputter-coated Platinum surfaces. Interestingly, they could show that in comparison to a glass surface, a thin layer of Pt (5 nm) did not exhibit any antibacterial effect [33]. Such a thin layer would probably not be sufficient to observe antibacterial effects.

In contrast, we used a coating thickness of 20 nm and observed a significant growth reduction with gold, ruthenium and tantalum coating. On the other hand, gold was used as a double layer with a ruthenium underlayer as a bonding agent. Others have shown that double layers produced by magnetron sputtering of Ag/Cu and Au/Cu were efficiently killing bacteria in the surrounding growth medium [6]. These authors conclude that the electrode potential of copper is less than that of silver and gold; thus, the copper surface could act as a sacrificial anode. In addition, these authors use an assay similar to the ASTM E2149-20 (Standard Test Method for Determining the Antimicrobial Activity of Antimicrobial Agents under Dynamic Contact Conditions). In this assay, bacteria are incubated in a buffer (not in growth medium) for three hours. Sometimes, this procedure leads to significant bacterial death for more than one log rate under control conditions. Induction of bacterial cell death under these conditions is dependent on copper release [6]. Copper ion release is not desired in every application, since these are toxic for eukaryotic cells also; thus, the United States Environmental Protection Agency (2013) set the maximum contaminant level goals for Cu at 1.3 mg/L [34]. Surprisingly, in our study, a double coat of relatively inert materials, such as ruthenium and gold, had a strong antibacterial effect directly on the surface, which might be useful when surfaces should not be considerably leaching ions, such as coatings for surgical instruments.

Surface roughness is one of the parameters that can be optimized to prevent bacterial growth, but this topic is still a matter of debate. Many studies claim that the smoother a surface is, the lower the chance of bacterial adhesion [34]. In our study, we cannot confirm this assumption because we can reveal a much higher biofilm formation on glass with an RRMS of 174 pm compared to gold with an RRMS of 246 pm. Taylor and coworkers showed that bacterial adhesion on surfaces for P. aeruginosa is strongest for a roughness that has features in the same dimension as the bacterial size (1–2 μm). Roughness values outside this optimal range lead to a lower bacterial adhesion [35]. This could be explained by the reduction in shear forces, which bacteria receive, when they are in these niches. In our study, roughness was not an essential factor for bacterial adhesion and survival.

Gold is one of the most inert metals of all metallic elements, but on a nanoscale level, gold is a catalytic material [36]. This strongly depends on the preparation method and on the support material. Magnetron sputtering with argon plasma, as used in this study, induces an atomic flux of gold atoms onto the support glass surface. There, the atoms nucleate and grow to catalyst clusters. These nanoparticles are formed by low interfacial binding energies of gold and support. Recent measurements have shown that alcohols produced as bacterial metabolites, such as glycerol, might be converted to ROS by gold-mediated catalysis [37]. Furthermore, tantalum-coated surfaces also reduce the growth of Pseudomonades. Sputter-coated tantalum surfaces oxidate quickly when exposed to air and form a thin passive layer of tantalum pentoxide (Ta2O5) [38]. In this line, tantalum oxide can act as a catalyst [39]. Taken together, this study suggests that the development of industrial antibacterial surfaces is a feasible enterprise, especially when using double layers of inert materials, such as gold, on ruthenium coatings.

## 5. Conclusions

In this study, it was shown that coating glass with the transition metals gold, ruthenium and tantalum by magnetron sputtering can produce antibacterial surfaces, which are highly effective against the Gram-negative bacterium Pseudomonas aeruginosa. Antibacterial efficacy of metal surfaces was in the following order: gold > tantalum > ruthenium; this could be analyzed by the use of SEM, CLSM and CFU assays. A potential mechanism of cell death was also analyzed and revealed the same testimony. Namely, gold produces the highest amount of ROS on the surface.

## Figures and Tables

**Figure 1 microorganisms-10-01843-f001:**
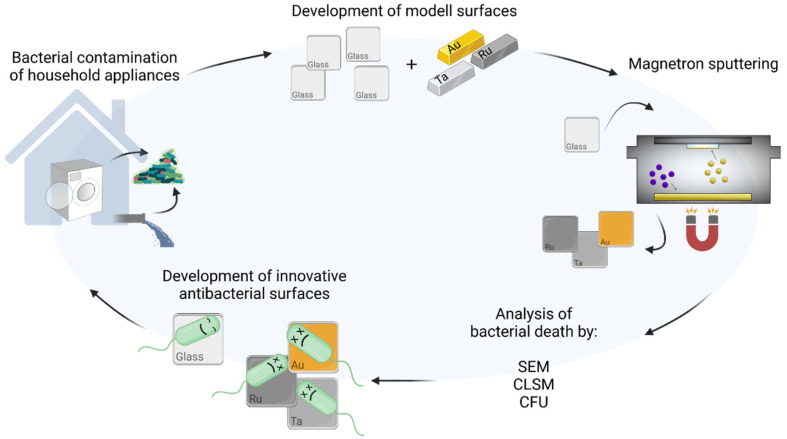
Scheme of the experimental design. On the left image, the problem of bacterial contamination in household appliances (such as washing machines or pipes) is depicted. The next step was to develop antibacterial metal surfaces by magnetron sputtering of gold, ruthenium and tantalum on glass slides (top and right images). These surfaces were then analyzed for antibacterial properties by SEM, CLSM and CFU (bottom images). While on glass slides bacterial growth was vital, metal-sputtered surfaces inhibited bacterial growth of Pseudomonas aeruginosa. Scheme was created with BioRender.com (accessed on 10 September 2022).

**Figure 2 microorganisms-10-01843-f002:**
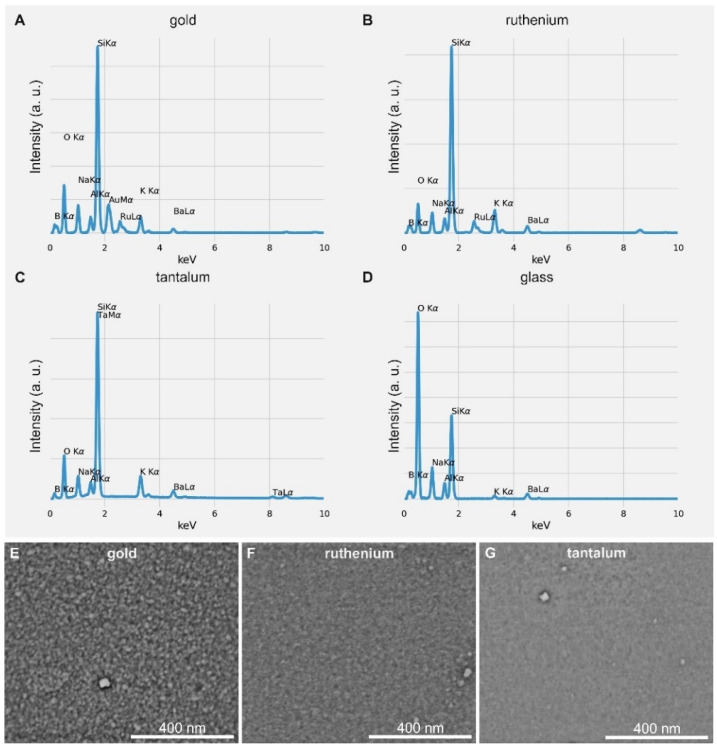
EDX spectra of three different metal coatings and pure glass, where (**A**–**C**) depict the EDX spectra of gold- (**A**), ruthenium- (**B**) and tantalum- (**C**) coated glass and (**D**) depicts the EDX spectrum of the uncoated glass surface. SEM images of the three different metal coatings, where (**E**) shows the gold surface, (**F**) the ruthenium surface and (**G**) the tantalum surface.

**Figure 3 microorganisms-10-01843-f003:**
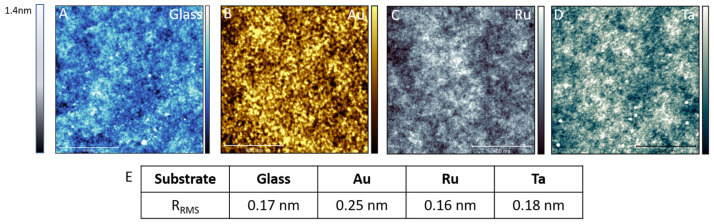
Analysis of surface roughness by atomic force microscopy of four different surfaces. As a control surface we used glass slides (**A**). Different metal coatings are shown in (**B**–**D**). (**B**) gold, (**C**) ruthenium and (**D**) tantalum. The scale bars in the images are 400 nm and the z height is 1.4 nm. In (**E**) the root mean square roughness for the different surfaces is shown.

**Figure 4 microorganisms-10-01843-f004:**
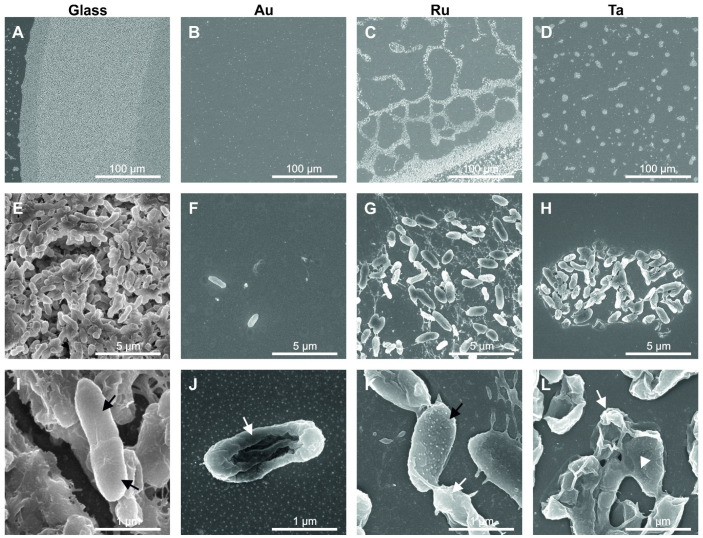
SEM images of Pseudomonas aeruginosa grown for 24 h on glass, gold, ruthenium and tantalum surfaces. The first row (**A**–**D**) shows an overview of bacterial growth on the different surfaces. On the glass control surface a vital bacterial biofilm is formed (**A**), whereas on the metal-coated surfaces a clear growth inhibition is visible (**C**,**D**). In the second row (**E**–**H**) a higher magnification is shown, revealing a healthy horizontal biofilm on glass (**E**) in contrast to single-spaced-out bacteria on gold (**F**) and remains of bacterial groups on ruthenium (**G**) and tantalum (**H**). High-resolution images are depicted in the last row (**I**–**L**). In the vital biofilm on glass a dividing bacterium is highlighted by black arrows (**I**). Metal coatings induced bacterial death, see white arrows (**J**–**L**). A biofilm slime residue is displayed by a white arrowhead on tantalum ((**L**), white arrowhead).

**Figure 5 microorganisms-10-01843-f005:**
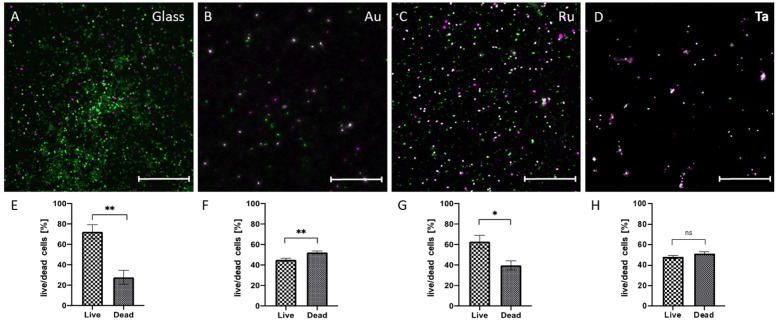
LIVE/DEAD Assay of Pseudomonas aeruginosa analyzed by CLSM. After cultivation of P. aeruginosa for 24 h on different surfaces bacteria were stained with Syto 9 representing live cells in green and dead cells in purple (propidium iodide) (**A**–**D**). Scale Bars = 20 µm. Evaluation of live and dead bacteria in percent (**E**–**H**). Note the high amounts of living bacteria on glass (**A**,**E**) in contrast to the highest amount of dead bacteria on gold (**B**,**F**). Definition of statistical significance: *p* < 0.05 (*), *p* < 0.01 (**), *p* < 0.001 (***), *p* < 0.0001 (****).

**Figure 6 microorganisms-10-01843-f006:**
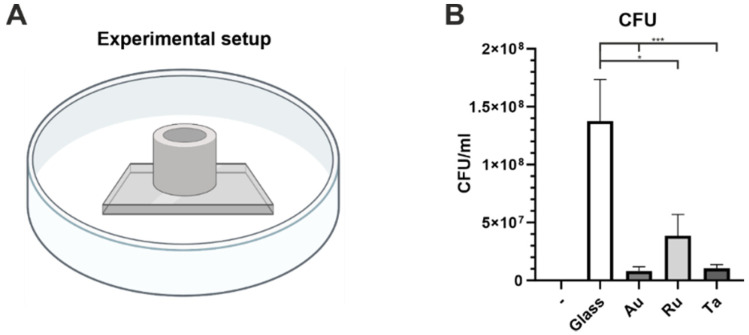
Colony Forming unit assay of Pseudomonas aeruginosa on different metal coatings and uncoated glass as control. On the left side a scheme of the experimental setup is depicted, showing a cloning cylinder on a glass slide in a Petri dish (**A**). This setup was used to ensure that the bacterial growth occurs only on the coated surface and in a defined area. On the right side a graph of the CFU assay per ml is presented (**B**). The first bar (-) is a result of a sterility control with glass incubated in LB medium only. Note that no colonies grow. Uncoated glass shows highly efficient bacterial growth (white-colored bar). Gold coating vastly reduced bacterial growth (medium gray bar). Ruthenium coating showed less antibacterial potential (light gray bar). Tantalum coating reduced bacterial growth nearly as efficiently as gold. Scheme in (**A**) was created with BioRender.com (accessed on 10 September 2022). Definition of statistical significance: *p* < 0.05 (*), *p* < 0.01 (**), *p* < 0.001 (***), *p* < 0.0001 (****).

**Figure 7 microorganisms-10-01843-f007:**
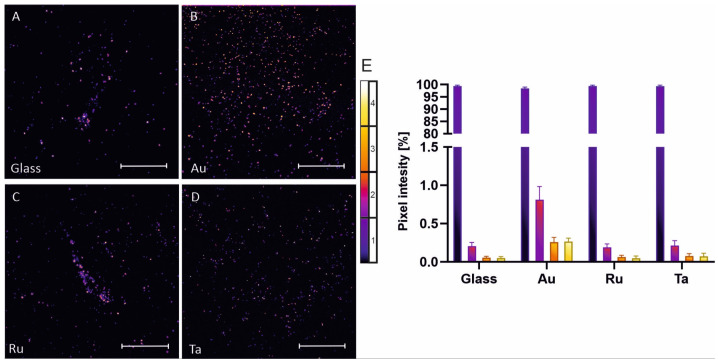
ROS assay of Pseudomonas aeruginosa grown for 24 h on glass, gold, ruthenium and tantalum surfaces. Reactive oxygen species were measured with the help of 2-7-dichlorofluorescin diacetate assay. Images were obtained by confocal laser scanning microscopy and are presented as false color images, using a fire ice false color coding (**A**–**D**). Highest intensity is depicted as yellow white and darkest pixels are shown as black blue. Most pixel values are in the background range (0–63 black blue) (**E**), while gold shows the highest amount of ROS production (orange and yellow bars).

## Data Availability

All presented data generated by our experiments are included in the manuscript.

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
