# Peer review of "Magnetron Sputtering of Transition Metals as an Alternative Production Means for Antibacterial Surfaces"

_microorganisms, 2022, doi:10.3390/microorganisms10091843_

Round 1

Reviewer 1 Report

The work deals with a current issue. The methodological approach is well chosen and the conclusions are conclusive. One thing that should be clarified: What is the reason for the much higher oxygen peak in the EDX spectrum (Fig 2D) on the pure glass substrate compared to the spectra with the very thin coatings? (and: put a blank between numbes and units, where its not done)

Reviewer 2 Report

This is the review of the manuscript microorganisms-1922257, namely "Magnetron sputtering of transition metals as an alternative production of antibacterial surfaces" by Bernhard Peter Kaltschmidt, Ehsan Asghari, Annika Kiel, Julian Cremer, Dario Anselmetti, Christian Kaltschmidt, Barbara Kaltschmidt, and Andreas Hütten 

General Comments:

In present work used bacterium to analyze the antibacterial effect on sputter coated surfaces. To detect bacterial cell death several assays were used such as scanning electron microscopy (SEM), LIVE/DEAD assays with confocal laser scanning microscopy (CLSM) and colony forming unit (CFU) assays. On glass bacterial growth is vivid depicted as “happy” bacteria, whereas coating with transition metals is lethal and leads to dying bacteria depicted as “unhappy” bacteria. Strongest growth inhibition was observed on gold and tantalum sputter-coated surfaces. These observations might pave a way for magnetron sputter coated gold surfaces of household appliances and/or surgical instruments to avoid unhealthy bacterial growth. The results of this study could be translated to the use of innovative antibacterial surfaces, which might lead to a lower consumption of antimicrobial chemicals in household appliances, thus protecting the environment.

The scientific research looks very good. I accept manuscript in present form.
